# Theoretical and Experimental Study on the Impact of Long-Wave Radiation on the Greenhouse Effect of a Prefabricated Temporary House

**DOI:** 10.3390/e24101446

**Published:** 2022-10-11

**Authors:** Qian Wen, Enshen Long

**Affiliations:** 1Institute for Disaster Management Reconstructions, Sichuan University, Chengdu 610065, China; 2College of Architecture and Environment, Sichuan University, Chengdu 610065, China

**Keywords:** theoretical and experimental study, long-wave radiation, greenhouse effect, prefabricated temporary house

## Abstract

In this paper, an experimental rig of a prefabricated temporary house (PTH) was first established. Then, predicted models for the thermal environment of the PTH with and without considering long-wave radiation were developed. Next, the exterior-surface, interior-surface and indoor temperatures of the PTH were calculated by using the predicted models. The calculated results were then compared with the experimental results to study the influence of long-wave radiation on the predicted characteristic temperature of the PTH. Finally, the predicted models were used to calculate the cumulative annual hours and the intensity of the greenhouse effect of four different climate cities (Harbin, Beijing, Chengdu, Guangzhou, China). The results showed that: (1) the predicted temperature values of the model considering long-wave radiation were closer to the experimental results; (2) the effect level of the long-wave radiation on the three characteristic temperatures of the PTH from big to small was: exterior-surface temperature, interior-surface temperature, and indoor temperature; (3) the long-wave radiation had the greatest impact on the predicted temperature value of the roof; (4) under different climate conditions, the cumulative annual hours and the intensity of the greenhouse effect considering long-wave radiation were smaller than those without considering long-wave radiation; (5) the duration of the greenhouse effect considering and ignoring long-wave radiation varied significantly with the climate region, and that in Guangzhou was the longest, followed by Beijing and Chengdu, and that in Harbin was the shortest.

## 1. Introduction

Prefabricated temporary houses (PTHs) have been widely used for resettling victims after disasters owing to their advantages such as convenient transportation, easy installation, and being used without electrical and thermal control systems installed [1,2]. However, the envelope of PTHs is normally made of lightweight materials and thinner than that of conventional buildings, thus, their insulation performance is relatively poor [3]. To some extent, it is intolerably hot in summer and cold in winter inside these PTHs [4].

In order to improve the indoor thermal environment of PTHs, so far, many scholars have performed massive relevant investigations with respect to long-wave radiation and/or other impact factors. Huang et al. [5] conducted a field test of a PTH, and the results showed that the roof interior-surface temperature of the PTH was 8.1 °C higher than the outdoor temperature, which meant that the greenhouse effect was generated inside the PTH in summer. Wang et al. [6] also carried out a field test by using PTHs with two layers in winter. The results indicated that the interior-surface temperature was slightly lower than the outdoor temperature, which meant that a cold-house effect was produced inside the PTH in winter. Wang et al. [7] made two different designs of PTH, and the results showed that one of the PTH was more suitable for disaster relief and outdoor low air environment. Wang et al. [8] reported an experimental study on the indoor thermal environment in a subtropical experimental PTH. The results showed that under a closed room environment, the PTH temperature was high in summer and low in winter, and the indoor temperature of the PTH was sensitive to the influence of height, and appropriate measures should be taken to improve the thermal environment of PTHs. Chen et al. [9] studied the indoor thermal environment of a PTH in winter considering the outdoor temperature, radiation, and other factors. The results showed that the indoor air temperature of the PTH in winter was low and the insulation performance was poor. Long et al. [10] simulated the annual heating and cooling energy consumption of four kinds of PTHs. The results showed that when the ventilation volume of the same building in the same city increased from 0 to 1.5, the annual heating energy consumption was greater than the annual cooling energy consumption. In addition, Long et al. [11] studied the influence of the same increase of the shape coefficient on the annual cooling and heating energy consumption of two PTHs under the climate conditions of 14 cities in China. Liu et al. [12] studied the trend of long-wave radiation in urban spaces surrounded by dense buildings, and a calculation model for long-wave radiation was established. They claimed that the rule of long-wave radiation in an enclosed space was obtained by analyzing different forms of building enclosure. Long et al. [13] conducted an experimental study on cooling and heating energy consumption in Tampa and Guangzhou. The results showed that the heating heat recovery and cooling heat recovery of different cities were similar when the external window heat transfer coefficient was the same. Long et al. [14] also carried out an envelope transformation of two more PTHs with the same method and found that local climate conditions were very important for choosing which energy saving measures to take in different places. Li et al. [15] proposed a residential building model and developed a simplified second-order lumped capacity system to study the dynamic thermal process of an indoor environment. The results showed that choosing the correct enclosure structure and heating load could reduce the heating energy consumption. Meng et al. [16] established a three-dimensional wall heat transfer model considering the thermal bridge effect of mortar joints and verified it with the thermoelectric analogy theory. The results showed that the position of the thermocouple and heat flux meter, the size and shape of the heat flux meter and the layout of pasting angle had a great influence on improving the measurement accuracy. Wang et al. [17] studied the time lag (TL) and decrement factor (DF) for a hollow double glazing by numerical modeling. The results showed that when the transmittance was 0.1, TL and DF decreased by about 3–17% compared to TL and DF without considering the transmittance; and the transmittance effects were stronger in summer than in winter. Lhomme et al. [18] studied the accuracy of downward long-wave radiation for frost prediction models. The results showed that during nighttime hours, the formula yielded reasonably good estimates when the value of ratio s was replaced by its mean value calculated the previous day between 14 h and 16 h 30 min. Castro et al. [19] described a simple methodology for measurement and calculation, with a good accuracy for the average atmospheric long-wave down-welling radiation using a tilted, low-cost infrared thermometer and tilt setting. The results showed that the divergence and radiation intensity between instantaneous data pairs depended on the asymmetry of cloud density. Dai et al. [20] used regression optimization software to evaluate the atmospheric radiation that had the greatest impact on the thermal characteristics of air conditioning and proposed a new atmospheric radiation model based on the atmospheric radiation database of seven stations below 2373 m altitude. Compared with the existing models, the new model had a higher accuracy, and the prediction results were in good agreement with the MODTRAN calculation results at different altitudes. Matzarakis et al. [21] evaluated six existing models of downward long-wave clear-sky irradiation using multiyear datasets recorded by the Regional KLIma Project in southwestern Germany. It gave estimates that were much closer to the measurements (within 5% in the lowlands and 7% in the mountains). Aubinet et al. [22] presented new empirical models for predicting daily mean heat radiation on sunny and cloudy days. Their biggest advantage was that they used only three variables: air temperature, water vapor pressure, and clarity index. In particular, neither cloud cover measurements nor temperature or humidity profiles were required. Niemela et al. [23] compared the results of several long-wave downlink radiation flux parameterization and hourly mean point surface radiation observations made in 1997 and 1999 in Sodankyla, Finland. It was found that almost all long-wave schemes generally underestimated the downwelling clear-air flux, especially under cold (surface inversion) conditions. Atwater et al. [24] studied the effect of atmospheric infrared radiation. The difference between surface temperature and sky temperature ranged from 5 °C to 20 °C and it was a complex function of season and geography. IR radiation is often parameterized by determining the equivalent sky temperature dependent on surface temperature. Notaridou et al. [25] calculated the downward flux of long-wave atmospheric radiation on the surface and its variation with height on sunny days and nights in Athens’ summer. The results showed that the values calculated by Idso and Jackson’s formula were in good agreement with those calculated by the model. Martin et al. [26] proposed a new algorithm to calculate the temperature of thermal radiation from the sky. The results of the calculations performed at 193 TMY sites in the continental United States were summarized. Therefore, radiative cooling of buildings seems to be a promising strategy for heat dissipation. Argiriou et al. [27] evaluated the radiative cooling potential of Athens using 12 years of hourly weather data to study the performance results of a simple radiator. The radiative cooling potential was determined by the ambient temperature, relative humidity, wind speed, and cloud cover, and a simple radiator could be used to estimate the cooling potential based on weather data from Athens. Hanif et al. [28] studied the relationship between radiative cooling power and temperature differences between the environment and the sky, taking the potential of radiative cooling systems in Malaysia as an example to evaluate. It was found that radiative cooling could save up to 11% of power consumption for cooling purposes. Zhao et al. [29] carried out theoretical analyses and field measurements to examine the long-wave infrared radiation properties of vertical green facades in the subtropical city of Guangzhou China. Based on the observation data, an empirical equation for transmitting long-wave infrared radiation was established. Moreover, a quantitative method was used to assess the accuracy of the long-wave infrared radiation model, and the results indicated that the calculated values were in good agreement with the measured values. Long et al. [30] proposed an idea of dynamic management of both solar radiation and long-wave thermal radiation. The results showed that the window with low emissivity could not reduce the energy consumption for cooling. The dual-intelligent window surpassed the traditional intelligent windows due to the fact that the application of the dual-intelligent window could reduce cooling energy by 21.7% compared with the traditional intelligent window.

However, no research presented models for the thermal environment of a PTH with and without considering long-wave radiation and evaluated its influence on the greenhouse and cold-house effect of the PTH. To bridge the gap, therefore, the rest of this paper consists of the following contents:An experimental rig for the PTH with detailed materials, sizes, and sensors is established first.Predicted models for thermal environment of PTH with and without considering long-wave radiation are then presented.The exterior-surface, interior-surface, and indoor temperatures of the PTH used in the experiment are calculated by using the predicted models with and without considering the long-wave radiation.The calculated results are compared with the experimental results to study the influence of long-wave radiation on the characteristic temperature of the PTH.The predicted models are used to calculate the cumulative annual hours and the intensity of the greenhouse effect of four different climate cities (Harbin, Beijing, Chengdu, Guangzhou, China).

## 2. Description of the Experimental System

The schematic of the experimental rig for the PTH is shown in Figure 1, and the materials and sizes of the roof, wall, floor, and windows of the PTH are presented in Table 1. In addition, the physical and thermal properties of these materials are given in Table 2.

The experimental PTH as displayed in Figure 2 was placed on the rooftop of a four-story building on the campus of Sichuan University in China, and the campus is located in the hot-summer and cold-winter zone. There were no high-rises to shade the PTH and no electrical and thermal equipment inside the PTH. The door and windows of the PTH were closed during the experiments.

Moreover, sensors utilized in the test are presented in Table 3. All the T-type thermocouples were calibrated before they were used in the experiments and were connected to a data logger, where all the measured temperatures were recorded at an interval of 5 min, because 5 min was long enough to collect stable measured data. In addition, the distribution of the temperature sensors is presented in Figure 2; for example, different temperature sensors were deployed on the exterior surface of the PTH as shown in the figure; to be specific, only one thermocouple was fixed on each side.

## 3. Predicted Models for Thermal Environment of PTH with and without Considering Long-Wave Radiation

The exterior-surface temperature of the PTH was subject to the atmospheric temperature, solar radiation, ground surface temperature, as well as the long-wave radiation from the sky. Indoor and interior-surface temperatures changed with the exterior-surface temperature of the PTH. According to the thermal balance of the PTH, at an interval time of Δτ, the total heat gain of the indoor air without considering long-wave radiation can be derived as below:(1)$$Q_{t}(\Delta \tau )=(Q_{body}+Q_{glass}+Q_{air})\Delta \tau$$
where
$$\left\{\begin{array}{l} Q_{t}(\Delta \tau )=c\;\rho V(t_{\tau }-t_{\tau -1})\\ Q_{body}={\;}_{\sum }K_{i}F_{i}\left(t_{outdoor}+\frac{aI}{\alpha_{out}}-t_{\tau }\right)\\ Q_{glass}={\;}_{\sum }F_{Gi}I(\eta_{i}+\frac{\alpha_{in}}{\alpha_{out}}\rho_{G})C_{i}\\ Q_{air}=\frac{n_{k}cV\rho (t_{outdoor}-t_{\tau })}{3600} \end{array}\right.$$
where *Q_t_* is the heat gain of the indoor air at a duration of τ; *Q_body_* is the heat gain from the envelope; *Q_glass_* is the heat gain from solar radiation through glass; *Q_air_* is the heat gain from air leakage; *t_τ_* is the indoor temperature at time τ in °C; *F_Gi_* is the transmitting area of glass window in m^2^; *I* is the solar radiation in W/m^2^; *η_i_* is the transmitting coefficient of solar radiation through the glass; *ρ_G_* is the absorbing coefficient of glass to solar radiation; *C_i_* is the shading coefficient; *α_in_* is the comprehensive heat transfer coefficient of air on the interior surface in W/m^2^·°C; *α_out_* is the comprehensive heat transfer coefficient of air on the exterior surface in W/m^2^·°C; *n_k_* is the air change frequency in time/h; *V* is the PTH volume in m^3^; *t_outdoor_* is the outdoor temperature in °C; *K _i_* is the thermal conductivity of envelope in W/m^2^·K; *F_i_* is the thermal area of the envelope in m^2^; *ρ* is the air density in kg/m^3^; *c* is the air specific heat in kJ/kg·°C; *t_τ_*_−1_ is the indoor temperature of the last instant time τ − 1 °C; and *a* is the solar radiation absorption factor.

By solving Equation (1), the indoor temperature at an instant time of τ without considering long-wave radiation can be deduced as follows:(2)$$t_{\tau }=\frac{\rho cVt_{\tau -1}+\left[\sum F_{Gi}I(\eta_{i}+\frac{\alpha_{in}}{\alpha_{out}}\rho_{G})C_{i}+\frac{n_{k}cV\rho t_{outdoor}}{3600}+\sum K_{i}F_{i}(t_{outdoor}+\frac{aI}{\alpha_{out}})\right]\Delta \tau }{\rho cV+(\frac{n_{k}cV\rho t_{outdoor}}{3600}+\sum K_{i}F_{i})\Delta \tau }$$
where *α_in_ =* 8.7 W/m^2^·°C; *α_out_ =* 23.3 W/m^2^·°C; *η_i_ =* 0.837; and *ρ_G_ =* 0.06.

Based on the Equation (2), an optimized model with respect to the influence of long-wave radiation on the PTH was further derived as follows to calculate the indoor temperature at time *τ*:(3)$$t_{\tau }=\frac{\rho cVt_{\tau -1}+\left[\sum F_{Gi}I(\eta_{i}+\frac{\alpha_{in}}{\alpha_{out}}\rho_{G})C_{i}+\frac{n_{k}cV\rho t_{outdoor}}{3600}+\sum K_{i}F_{i}(t_{outdoor}+\frac{aI}{\alpha_{out}}-\frac{Q_{{}_{lw}}}{\alpha_{out}})\right]\Delta \tau }{\rho cV+(\frac{n_{k}cV\rho t_{outdoor}}{3600}+\sum K_{i}F_{i})\Delta \tau }$$
where *Q_lw_* is the long-wave radiation in W/m^2^, which can be written as below:(4)$$Q_{lw}=\left\{\begin{array}{ll} \sigma \varepsilon_{w}T_{exterior}^{4}-\varepsilon_{w}q_{ADR} & \;\mathrm{horizontal}\;\\ \frac{1}{2}(\sigma \varepsilon_{w}T_{exterior}^{4}-\varepsilon_{w}q_{ADR})+\frac{1}{2}(\sigma \varepsilon_{w}\varepsilon_{g}T_{exterior}^{4}-\sigma \varepsilon_{w}\varepsilon_{g}T_{g}^{4}) & \mathrm{vertical}\; \end{array}\right.$$
where *q_ADR_* is the atmospheric downward radiation in W/m^2^, *σ* is the Stefan–Boltzmann constant, *σ* = 5.67 × 10^−8^ W/(m^2^·K^4^); *ε_W_* is Stephan-Boltzmann constant emissivity coefficient of exterior-surface; *ε_g_* is the emissivity coefficient of the ground; *T_exterior_* is the temperature of the exterior surface in K; and *T_g_* is the temperature of the ground in K.

The interior-surface and exterior-surface temperatures were calculated according to the heat balance of the PTH envelope as below:(5)$$\left\{\begin{array}{l} aI-Q_{lw}+\alpha_{out}(t_{outdoor}-t_{exterior})+\frac{\left(t_{interior}-t_{exterior}\right)}{R}=0\\ q_{rin}+\alpha_{in}(t_{\tau }-t_{inner})+\frac{\left(t_{exterior}-t_{interior}\right)}{R}=0 \end{array}\right.$$
where *q_rin_* is the heat gain from direct solar radiation through glass in W/m^2^; *t_inner_* is the interior-surface temperature in °C; and *t_exterior_* is the exterior-surface temperature in °C. An initial value for the indoor temperature at time τ−1 was assigned, which was the measured outdoor temperature, and the indoor, interior-surface, and exterior-surface temperatures at time τ could be solved with Equations (3) and (5).

## 4. Results and Discussion

The exterior-surface, interior-surface, and indoor temperatures of the PTH used in the experiment were calculated by using the predicted models with and without considering the long-wave radiation. Then, the calculated results were compared with the experimental results to study the influence of long-wave radiation on the characteristic temperature of the PTH, and to reveal the influence of long-wave radiation on the greenhouse and cold-house effects of the experimental PTH. Finally, the predicted models were used to calculate the cumulative annual hours and intensity of the greenhouse effect of four different climate cities (Harbin, Beijing, Chengdu, Guangzhou, China), so as to analyze the effect of long-wave radiation on the greenhouse effect of a PTH under different climate conditions.

### 4.1. Exterior-Surface Temperature of the PTH

#### 4.1.1. Summer

Figure 3 shows the model-predicted temperature value with and without considering the long-wave radiation and measured temperature value of five exterior surfaces (south, west, east, north and roof) of the PTH during a summer (0:00 of 23 July–0:00 of 28 July). Table 4 displays the average difference between the two predicted values of each exterior surface during the same period, the maximum difference between the two predicted values of each exterior surface at night, and the maximum difference between each exterior surface in the daytime during the same period. It can be seen that the average difference between the two predicted values of the roof were larger than those of the other surfaces, that is to say, the long-wave radiation had the most obvious influence on the predicted exterior-surface temperature of the roof. The average difference between the two predicted values of the remaining surfaces was not obvious, in comparison, and the average difference of the two predicted values on the northern surface was the least. Moreover, for all exterior surfaces, the maximum difference between the two predicted values in the daytime was larger than that at night during the period; therefore, the influence of long-wave radiation on the predicted value of the exterior-surface temperature in the daytime was greater than that at night in summer.

In order to evaluate the difference between the two predicted temperatures and the measured temperature, the root-mean-square error (RMSE) and coefficient variable of the root-mean-square error (CV) were used to evaluate the difference between them as given below:(6)$$RMSE=\sqrt{\sum_{i=1}^{N}{\left(m_{i}-s_{i}\right)}^{2}/N}$$
(7)$$CV(\%)=\frac{RMSE}{m_{ave}}$$
where *m_i_* is the measured temperature at time *i*, *s_i_* is the predicted temperature at time *i*, *N* is the total time; and *m_ave_* is the average measured temperature. In general, when the CV is less than 30%, the predicted value is quite close to the measured temperature [31].

Table 5 illustrates the RMSE and CV between the two predicted values and the measured value during the period. As can be seen from Table 5, when ignoring the influence of long-wave radiation on the exterior-surface temperature, the RMSE between the predicted value and the measured value was 1.58–6.32 °C, and the CV was 5.81–17.32%. When considering the effect of long-wave radiation, the RMSE between the predicted value and measured value of the exterior-surface temperature was 1.46–2.88 °C, and the CV was 4.73–8.79%. Thus, the RMSE and CV between the predicted value and the measured value when considering long-wave radiation were smaller than those without considering long-wave radiation in all directions, that is, the predicted value considering long-wave radiation was closer to the measured value. By contrast, the RMSE and CV between the two predicted values of the roof were relatively large.

#### 4.1.2. Winter

Figure 4 presents the model-predicted temperature value with and without considering the long-wave radiation and measured temperature value of five exterior-surfaces (south, west, east, north and roof) of the PTH during a winter (0:00 of 13 February–0:00 of 17 February). Table 6 demonstrates the average difference between the two predicted values of each exterior surface during the same period, the maximum difference between the two predicted values of each exterior surface at night, and the maximum difference between each exterior surface in the daytime during the same period. It can be seen from Figure 4 and Table 6 that the average difference between the two predicted values of the roof was larger than that of the other surfaces no matter if it was night or daytime, that is to say, the long-wave radiation had the most observable influence on the predicted exterior-surface temperature of the roof. The average difference between the two predicted values of the rest surfaces was not obvious. Moreover, it can be seen that the maximum difference between the two predicted values in the daytime in winter was larger than that at night in winter for all exterior surfaces; therefore, the influence of long-wave radiation on the predicted value of exterior-surface temperature in the daytime in winter was also greater than that at night in winter.

Table 7 shows the RMSE and CV between the two predicted values and the measured value with and without considering the long-wave radiation during this period. As can be seen from Table 7, when ignoring the influence of long-wave radiation on the exterior-surface temperature, the RMSE between the predicted and measured value was 1.36–7.35 °C, and the CV was 9.92–51.41%. When considering the effect of long-wave radiation, the RMSE between the predicted and measured value of the exterior-surface temperature was 1.02–3.88 °C, and the CV was 7.74–27.14%. Thus, the RMSE and CV between the predicted value and the measured value when considering long-wave radiation were smaller than those without considering long-wave radiation in all directions, that is, the predicted value considering long-wave radiation in winter was closer to the measured value. By contrast, the RMSE and CV between the two predicted values of the roof were quite large. Furthermore, it can be found that, the measured temperature in the winter (or the denominator of Equation (7)) was evidently lower than that in the summer, thus, the CV in the winter was larger than that in the summer, as the RMSE in the summer or winter (the numerator of Equation (7)) exhibited no obvious difference.

According to the results of Section 4.1.1 and Section 4.1.2, it can be concluded that, whether in summer or winter, the long-wave radiation during the daytime had a greater impact on the predicted exterior-surface temperature than that at night. The long-wave radiation had the most significant effect on the predicted temperature value of the roof’s exterior-surface. The predicted value of the theoretical model considering long-wave radiation was closer to the measured value in all directions.

### 4.2. Interior-Surface Temperature of the PTH

#### 4.2.1. Summer

Figure 5 exhibits the model-predicted temperature value with and without considering the long-wave radiation and measured temperature value of five interior surfaces of the PTH during the same period in the summer. Table 8 reveals the average difference between the two predicted values of each interior surface during the same period, the maximum difference between the two predicted values of each interior surface at night, and the maximum difference between each interior surface in the daytime during the same period. It can be seen that the average difference between the two predicted values of the roof was larger than that of other surfaces no matter the time of day, namely, the long-wave radiation had the most observable influence on the predicted interior-surface temperature of the roof. The average difference between the two predicted values of the rest surfaces was not obvious; in comparison, the average difference of the two predicted values on the northern surface was the least. Moreover, the maximum difference between the two predicted values in the daytime in summer was larger than that at night in summer for all interior surfaces; therefore, the influence of long-wave radiation on the predicted value of interior-surface temperature in the daytime in summer was larger than that at night in summer.

Table 9 shows the RMSE and CV between the two predicted values and the measured value with and without considering the long-wave radiation during this period. It can be seen that when ignoring the influence of long-wave radiation on the interior-surface temperature, the RMSE between the predicted value and the measured value was 1.43–2.29 °C, and the CV was 4.52–6.94%. When considering the effect of long-wave radiation, the RMSE between the predicted value and measured value of the interior-surface temperature was 1.22–1.93 °C, and the CV was 3.84–5.90%. Therefore, the RMSE and CV between the predicted value and the measured value when considering long-wave radiation were smaller in all directions, that is, the predicted value considering long-wave radiation in summer was closer to the measured value. By contrast, the RMSE and CV between the two predicted values of the roof were relatively large.

Similar to the influence of long-wave radiation on the predicted value of exterior-surface temperature, the long-wave radiation had the most significant influence on the predicted value of the interior-surface temperature of the roof in summer. The effect of long-wave radiation on the predicted interior-surface temperature in the daytime in summer was more significant than that at night in summer. The predicted value of the interior-surface temperature considering long-wave radiation was closer to the measured value. In contrast, the effect of long-wave radiation on the predicted value of interior-surface temperature was less than that of exterior-surface temperature.

#### 4.2.2. Winter

Figure 6 depicts the model-predicted temperature value with and without considering the long-wave radiation and measured temperature value of five interior surfaces of the PTH during the same period of the winter. Table 10 illustrates the average difference between the two predicted values of each interior surface during the same period, the maximum difference between the two predicted values of each interior surface at night, and the maximum difference between each interior surface in the daytime during the same period. It can be seen that the long-wave radiation had the most observable influence on the predicted interior-surface temperature of the roof. The average difference between the two predicted values of the rest surfaces was not obvious; in comparison, the average difference of the two predicted values on the southern surface was the smallest. Moreover, the influence of long-wave radiation on the predicted value of interior-surface temperature in the daytime in winter was greater than that at night in winter.

Table 11 presents the RMSE and CV between the two predicted values and the measured value with and without considering the long-wave radiation during this period. It can be seen that when ignoring the influence of long-wave radiation on the interior-surface temperature, the RMSE between the predicted value and the measured value was 1.50–2.17 °C, and the CV was 10.47–20.97%. When considering the effect of long-wave radiation, the RMSE between the predicted value and measured value of the interior-surface temperature was 1.40–1.92 °C, and the CV was 9.54–15.91%. Therefore, the RMSE and CV between the predicted value and the measured value when considering long-wave radiation were smaller in all directions, that is, the predicted value considering long-wave radiation in winter was closer to the measured value. By contrast, the RMSE and CV between the two predicted values of the roof were relatively large.

The summary of Section 4.2. is as follows: in both summer and winter, similar to the results of the exterior-surface temperature, the daytime long-wave radiation had a more significant effect on the predicted interior-surface temperature than that at night, especially on the interior surface of the roof. The predicted value of the interior-surface temperature considering long-wave radiation was closer to the measured value. The effect of long-wave radiation on the predicted interior-surface temperature in summer was greater than that in winter. In both summer and winter, the effect of long-wave radiation on the predicted interior-surface temperature was less than that of long-wave radiation on the predicted exterior-surface temperature.

### 4.3. Indoor Temperature of the PTH

#### 4.3.1. Summer

Figure 7 exhibits the model-predicted indoor temperature with and without considering the long-wave radiation and measured indoor temperature of the PTH during the same period of the summer. It can be seen that the influence of long-wave radiation in the daytime on the indoor temperature was greater than that at night, but its influence on the predicted indoor temperature was weaker than its influence on the predicted temperature of the exterior surface and interior surface; therefore, the predicted temperature with and without considering the long-wave radiation was very close to the measured temperature.

#### 4.3.2. Winter

Figure 8 shows the model-predicted indoor temperature with and without considering the long-wave radiation and measured indoor temperature of the PTH during the same period of the winter. The results were similar to that of the summer, that is, the influence of long-wave radiation on the predicted indoor temperature was weaker than its influence on the predicted temperature of the exterior surface and interior surface; therefore, the predicted temperature with and without considering the long-wave radiation was very close to the measured temperature.

From Section 4.1, Section 4.2 and Section 4.3, it can be seen that the predicted value of the temperature considering long-wave radiation was closer to the measured value of the temperature. In the daytime of summer, there was a greenhouse effect phenomenon so that the indoor and interior-surface temperature of the PTH was higher than the outdoor temperature. In winter nights, the PTH exhibited the phenomenon of the cold-house effect, that is, the indoor- and interior-surface temperatures were lower than the outdoor temperature. Comparatively speaking, the greenhouse effect phenomenon was more obvious in the daytime in summer considering long-wave radiation; in this case, the indoor environment of the PTH was very bad.

As a result, in the following section, the predicted model ignoring and considering the long-wave radiation model was used to calculate the cumulative annual hours and the intensity of the greenhouse effect of four different cities (Harbin, Beijing, Chengdu, Guangzhou, China) in different climate regions, so as to analyze the effect of long-wave radiation under different climate conditions on the greenhouse effect.

### 4.4. Climate Difference in the Effects of Long-Wave Radiation on PTH Greenhouse Effect

Referring to the data of China Meteorological Data Service Centre [32], basic climate conditions of the four cities are presented in Figure 9 and Figure 10. Figure 9 depicts the hours of outdoor air temperature distribution for the four cities throughout the year. The cities with a number of hours with an outdoor air temperature less than 18 °C in descending order were: Harbin > Beijing > Chengdu > Guangzhou. The cities with a number of hours with an outdoor air temperature greater than 26 °C in descending order were: Guangzhou > Beijing > Chengdu > Harbin. The number of hours greater than 26 °C in summer in Chengdu was less than that in Beijing.

Figure 10 depicts the monthly mean outdoor temperature over time in the four cities. It can be seen that Guangzhou had the smallest monthly mean outdoor temperature difference between summer and winter, while Harbin had the largest monthly mean outdoor temperature difference between summer and winter. In summer, the maximum value of the monthly mean outdoor temperature occurred in Guangzhou, while the minimum value occurred in Harbin. In winter, the monthly average outdoor temperature in Guangzhou was higher than that in the other three cities, and Harbin had the lowest monthly average outdoor temperature. The order of the monthly mean outdoor temperatures in the four cities from largest to smallest was: Guangzhou > Chengdu > Beijing > Harbin.

#### 4.4.1. Severe Cold Region

Figure 11 depicts the cumulative annual hours of greenhouse effect by using the predicted model with and without considering long-wave radiation in Harbin City located in the severe cold region, when it was under solar radiation and the indoor temperature was greater than the outdoor air temperature (outdoor air temperature greater than 26 °C). It can be seen that the cumulative number of annual greenhouse effect hours when ignoring long-wave radiation was 942 h, while the cumulative number of annual greenhouse effect hours when considering long-wave radiation was 839 h. When the long-wave radiation was ignored, the external heat dissipation of the PTH through the exterior surface was ignored, and the calculated value of the indoor temperature was larger, so the number of cumulative hours of greenhouse effect was larger.

Figure 12 illustrates the annual greenhouse effect intensity calculated by the predicted model ignoring and considering long-wave radiation in Harbin City. When the long-wave radiation was neglected, the greenhouse effect intensity ranged 3.5–5.0 °C. When long-wave radiation was considered, the greenhouse effect intensity was in the range 2.5–4.5 °C. Therefore, the greenhouse effect intensity when considering long-wave radiation was smaller than that when ignoring long-wave radiation.

#### 4.4.2. Cold Region

Figure 13 displays the cumulative annual hours of greenhouse effect by using the predicted model with and without considering long-wave radiation in Beijing City located in the cold region, when it was also under solar radiation and the indoor temperature was greater than the outdoor air temperature (outdoor air temperature greater than 26 °C). It can be seen that the cumulative number of annual greenhouse effect hours when ignoring long-wave radiation was 1536 h, while the cumulative number of annual greenhouse effect hours when considering long-wave radiation was 1628 h, that is to say, the cumulative number of hours of greenhouse effect when considering long-wave radiation was smaller than that when ignoring long-wave radiation.

Figure 14 shows the annual greenhouse effect intensity calculated by the predicted model ignoring and considering long-wave radiation in Beijing City. When the long-wave radiation was considered, the greenhouse effect intensity ranged 3.0–5.0 °C. When long-wave radiation was neglected, the greenhouse effect intensity was in the range 3.5–6.0 °C. Thus, the greenhouse effect intensity when considering long-wave radiation was smaller than that when ignoring long-wave radiation.

#### 4.4.3. Hot-Summer and Cold-Winter Region

Figure 15 displays the cumulative number of annual hours of greenhouse effect by using the predicted model with and without considering long-wave radiation in Chengdu City located in the hot-summer and cold-winter region, when it was also under solar radiation and the indoor temperature was greater than outdoor air temperature (outdoor air temperature greater than 26 °C). It can be seen that the cumulative number of annual greenhouse effect hours when ignoring long-wave radiation was 1446 h, while the cumulative number of annual greenhouse effect hours when considering long-wave radiation was 1543 h, that is to say, the cumulative number of hours of greenhouse effect when considering long-wave radiation was shorter than that when ignoring long-wave radiation.

Figure 16 indicates the annual greenhouse effect intensity calculated by the predicted model ignoring and considering long-wave radiation in Chengdu City. When the long-wave radiation was considered, the greenhouse effect intensity ranged 3.0–5.0 °C. When long-wave radiation was neglected, the greenhouse effect intensity was in the range 3.5–6.0 °C. Hence, the greenhouse effect intensity when considering long-wave radiation was also smaller than that when ignoring long-wave radiation.

#### 4.4.4. Hot-Summer and Warm-Winter Region

Figure 17 illustrates the cumulative number of annual hours of greenhouse effect by using the predicted model with and without considering long-wave radiation in Guangzhou City, a city with a hot summer and warm winter. It can be seen that during the daytime from April to October, the number of hours of greenhouse effect of the PTH exceeded 200 h; during the daytime of July and August, the number of hours of greenhouse effect of the PTH reached 400 h. The greenhouse effect basically appeared in the daytime of summer. The cumulative number of annual greenhouse effect hours when considering long-wave radiation was 2768 h, while the cumulative number of annual greenhouse effect hours when ignoring long-wave radiation was 2858 h, so the cumulative number of hours when considering long-wave radiation was smaller than the cumulative number of hours when ignoring long-wave radiation.

Figure 18 indicates the annual greenhouse effect intensity calculated by the predicted model ignoring and considering long-wave radiation in Guangzhou City. When the long-wave radiation was considered, the greenhouse effect intensity ranged 3.0–5.0 °C. When long-wave radiation was neglected, the green-house effect intensity was in the range 3.5–6.0 °C. Hence, the greenhouse effect intensity when considering long-wave radiation was also smaller than that when ignoring long-wave radiation.

According to the results of Section 4.4, among the four cities with different climate conditions, the duration of the greenhouse effect in Guangzhou was the longest, followed by Beijing, and that in Harbin was the shortest. In terms of the greenhouse effect intensity of the four cities, it was relatively shorter in Harbin, and it was almost the same with the other three cities. Both the cumulative number of annual hours and the intensity of the greenhouse effect when considering long-wave radiation were smaller than those without considering long-wave radiation.

## 5. Conclusions

In this paper, three characteristic temperatures of the PTH calculated by using the predicted model with and without considering the long-wave radiation were firstly compared with the measured value. Then, the cumulative number of annual hours and the intensity of the greenhouse effect of cities in four typical climate regions were calculated by using the predicted model, and the climate difference of the influence of long-wave radiation on the PTH greenhouse effect was evaluated. The main findings were as follows:(1)In both summer and winter, the predicted temperature values of the model when considering long-wave radiation were closer to the experimental results than those of the model ignoring long-wave radiation.(2)The effect level of the long-wave radiation on the three characteristic temperatures of the PTH from big to small was: exterior-surface temperature, interior-surface temperature, and indoor temperature.(3)For the five surfaces, the long-wave radiation had the greatest impact on the predicted temperature value of the roof; therefore, it was one of the most effective ways to improve the thermal environment of the PTH from the perspective of the radiation absorption and emission and thermal insulation characteristics of the roof.(4)Under different climate conditions, the cumulative number of annual hours and the intensity of the greenhouse effect when considering long-wave radiation were smaller than those without considering long-wave radiation.(5)The duration of the greenhouse effect when considering and ignoring long-wave radiation varied significantly with the climate region, and that in Guangzhou was the longest, followed by Beijing and Chengdu, and that in Harbin was the shortest; under different climate conditions, there was almost no obvious difference between the greenhouse effect intensity except in Harbin.

## Figures and Tables

**Figure 1 entropy-24-01446-f001:**
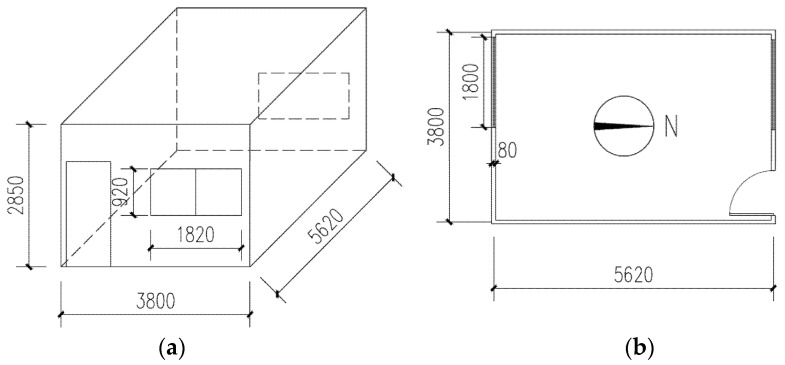
The schematic of the experimental PTH: (**a**) 3D schematic; (**b**) 2D schematic.

**Figure 2 entropy-24-01446-f002:**
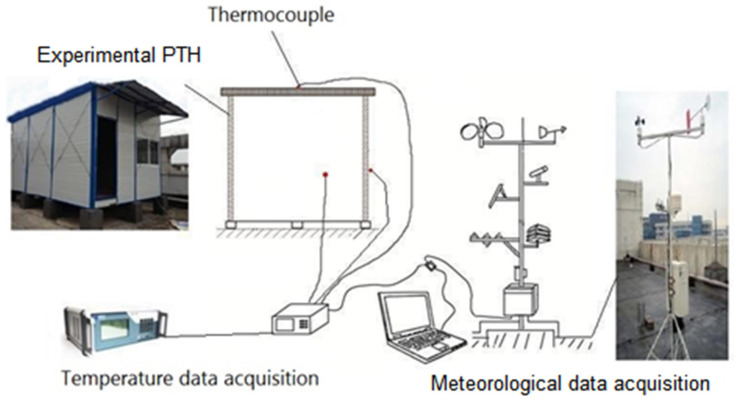
Schematic and photos of the experimental rig.

**Figure 3 entropy-24-01446-f003:**
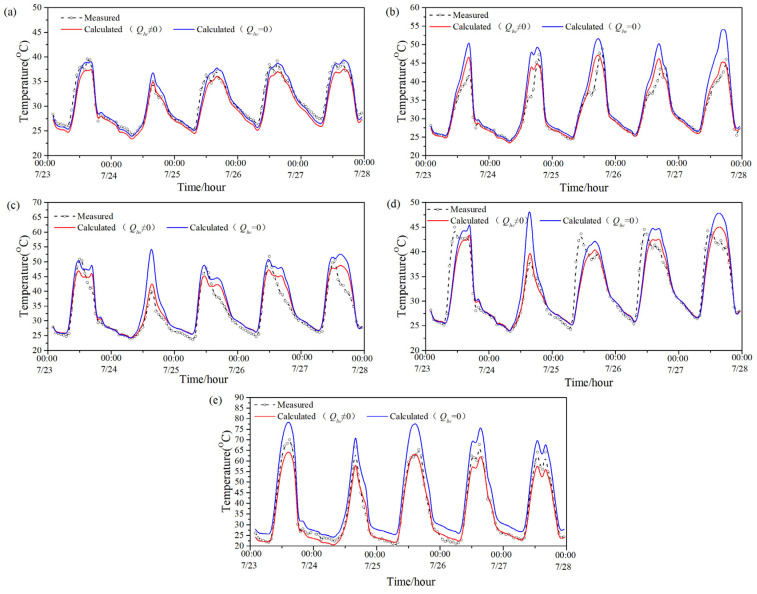
Two predicted values and measured value of the PTH’s exterior-surface temperature in summer: (**a**) south; (**b**) west; (**c**) east; (**d**) north; (**e**) roof.

**Figure 4 entropy-24-01446-f004:**
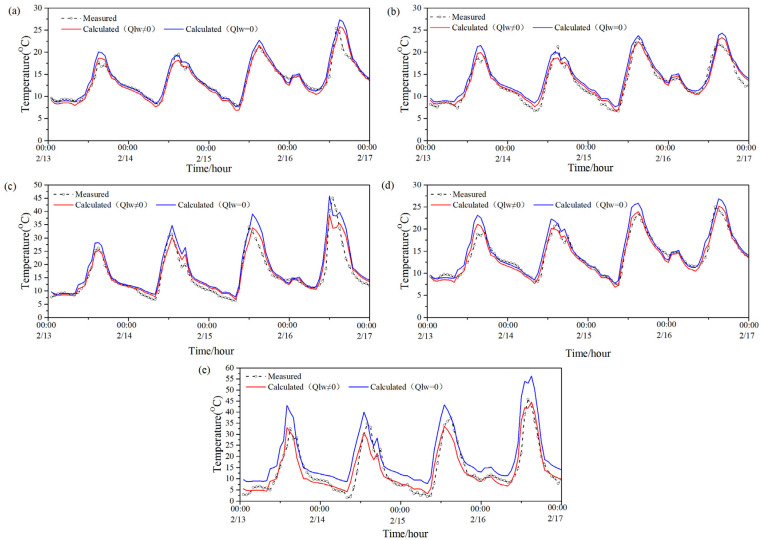
Two predicted values and measured value of the PTH’s exterior-surface temperature in winter: (**a**) south; (**b**) west; (**c**) east; (**d**) north; (**e**) roof.

**Figure 5 entropy-24-01446-f005:**
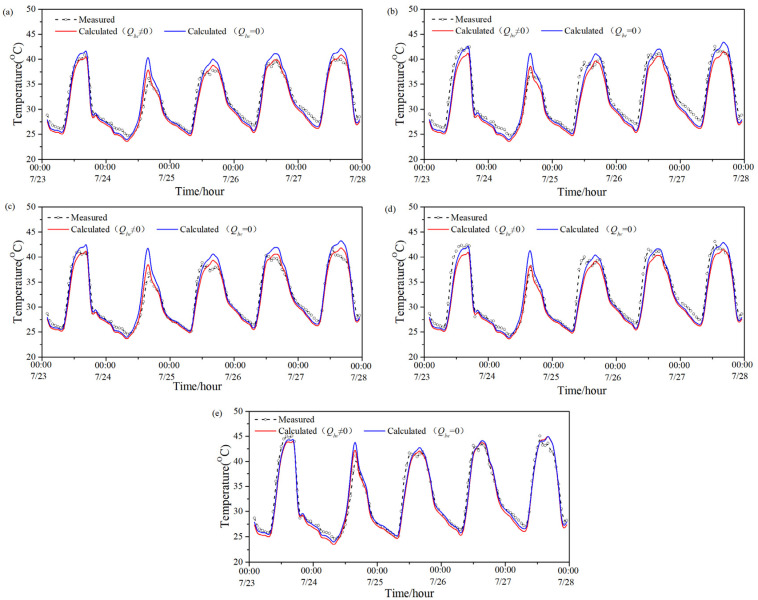
Two predicted values and measured value of the PTH’s interior-surface temperature in summer: (**a**) south; (**b**) west; (**c**) east; (**d**) north; (**e**) roof.

**Figure 6 entropy-24-01446-f006:**
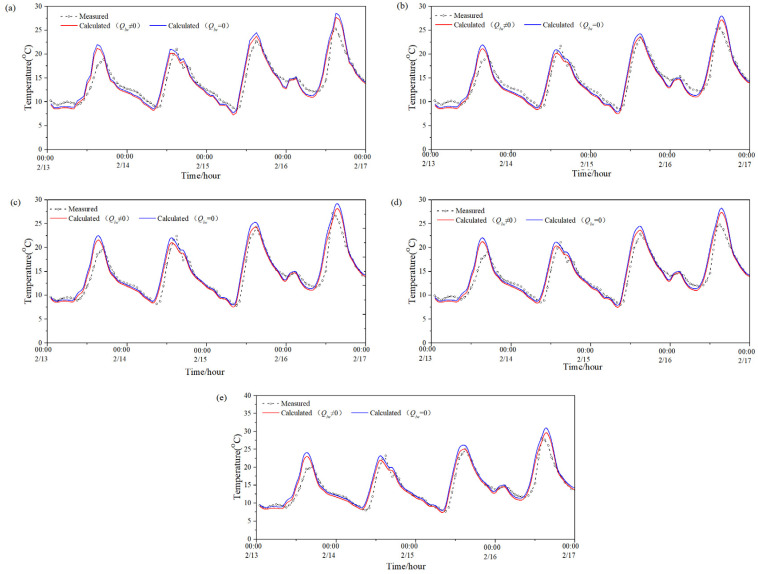
Two predicted values and measured value of the PTH interior-surface temperature in winter: (**a**) south; (**b**) west; (**c**) east; (**d**) north; (**e**) roof.

**Figure 7 entropy-24-01446-f007:**
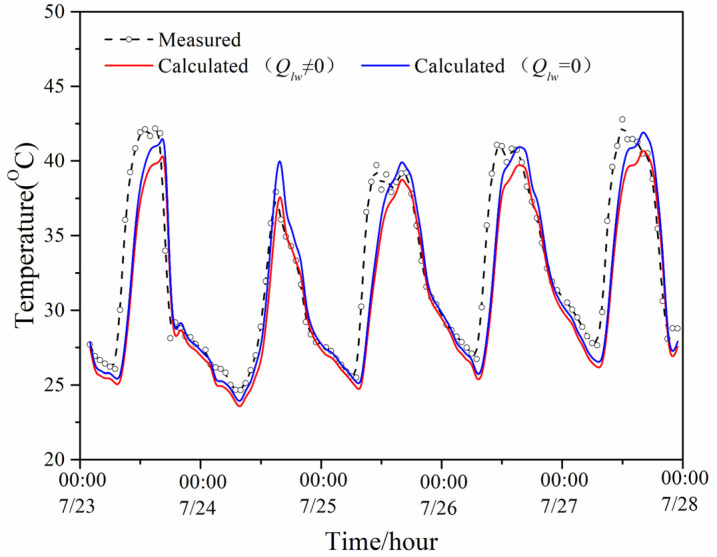
Two predicted values and measured value of the PTH indoor temperature in summer.

**Figure 8 entropy-24-01446-f008:**
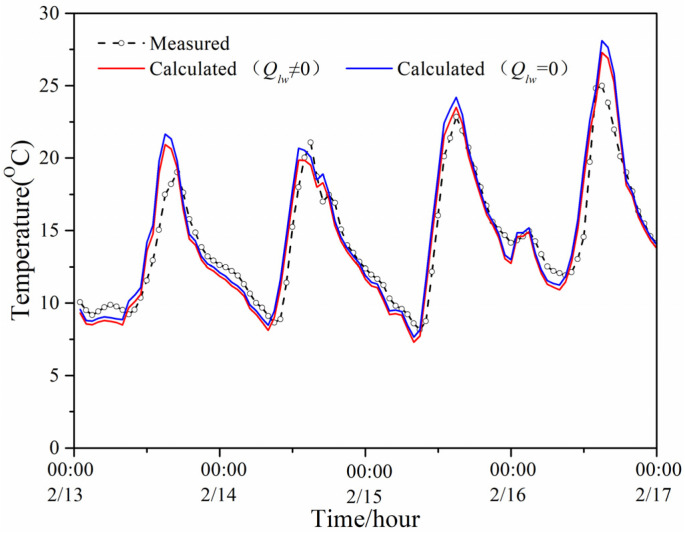
Two predicted values and measured value of the PTH indoor temperature in winter.

**Figure 9 entropy-24-01446-f009:**
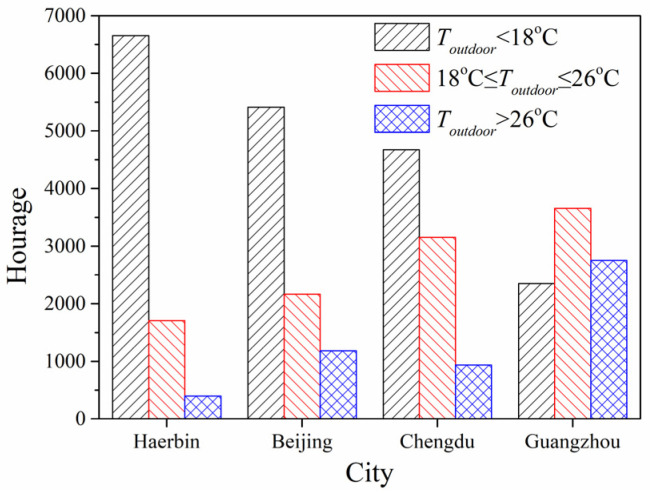
Hours of outdoor air temperature distribution in four cities.

**Figure 10 entropy-24-01446-f010:**
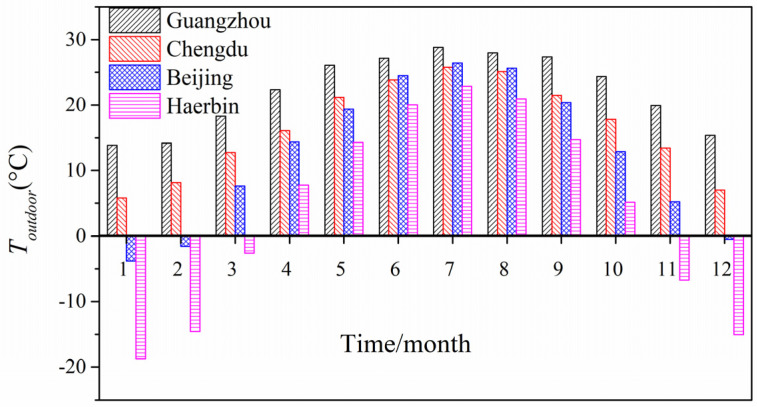
Average monthly outdoor air temperature in the four cities.

**Figure 11 entropy-24-01446-f011:**
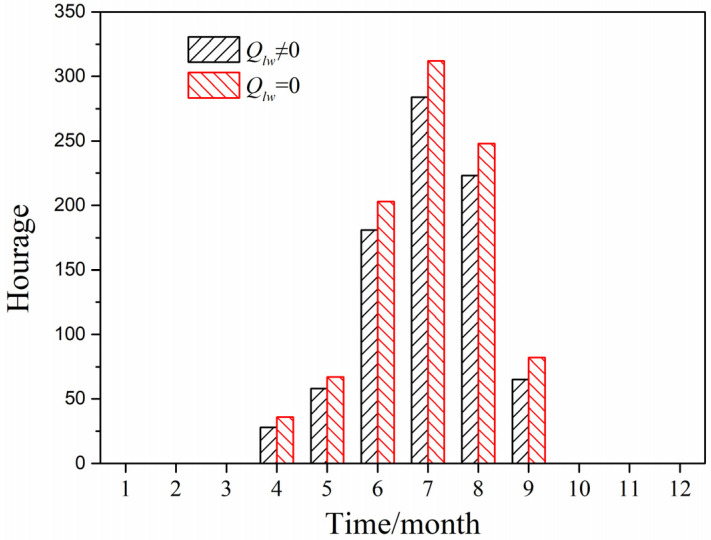
Cumulative hours of greenhouse effect in Harbin City.

**Figure 12 entropy-24-01446-f012:**
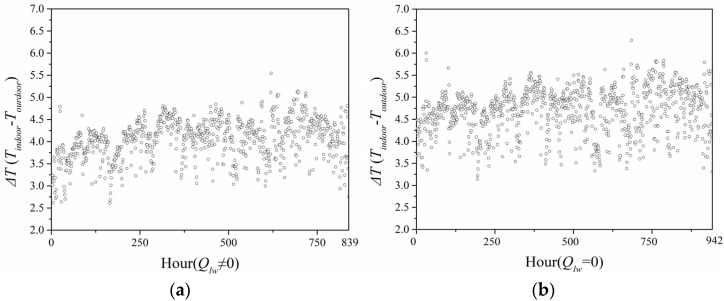
Intensity of greenhouse effect in Harbin City: (**a**) considering long-wave radiation; (**b**) neglecting long-wave radiation.

**Figure 13 entropy-24-01446-f013:**
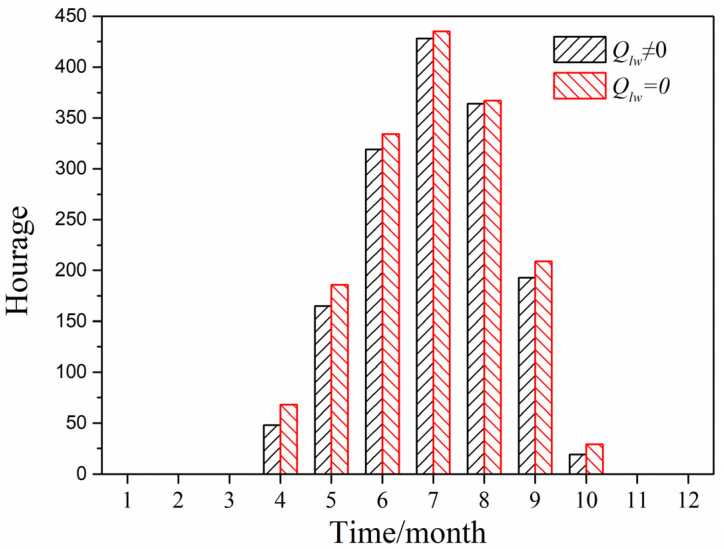
Cumulative hours of greenhouse effect in Beijing City.

**Figure 14 entropy-24-01446-f014:**
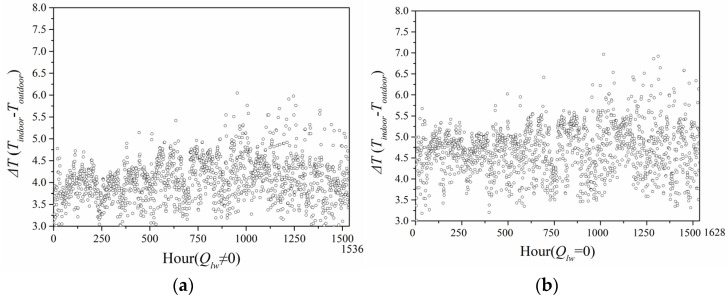
Intensity of greenhouse effect in Beijing City: (**a**) considering long-wave radiation; (**b**) neglecting long-wave radiation.

**Figure 15 entropy-24-01446-f015:**
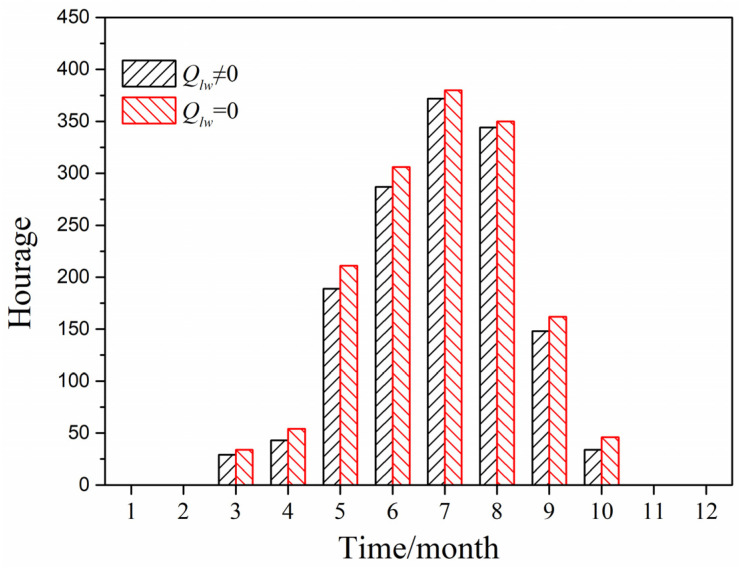
Cumulative hours of greenhouse effect in Chengdu City.

**Figure 16 entropy-24-01446-f016:**
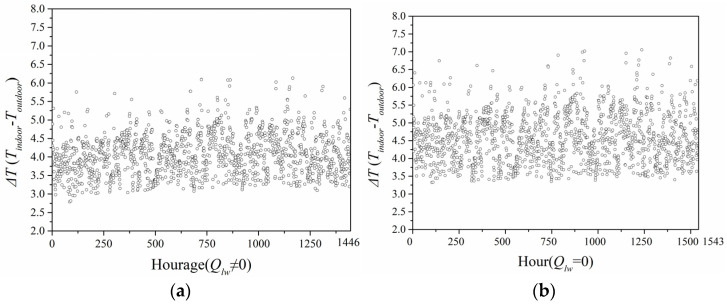
Intensity of green-house effect in Chengdu City: (**a**) considering long-wave radiation; (**b**) neglecting long-wave radiation.

**Figure 17 entropy-24-01446-f017:**
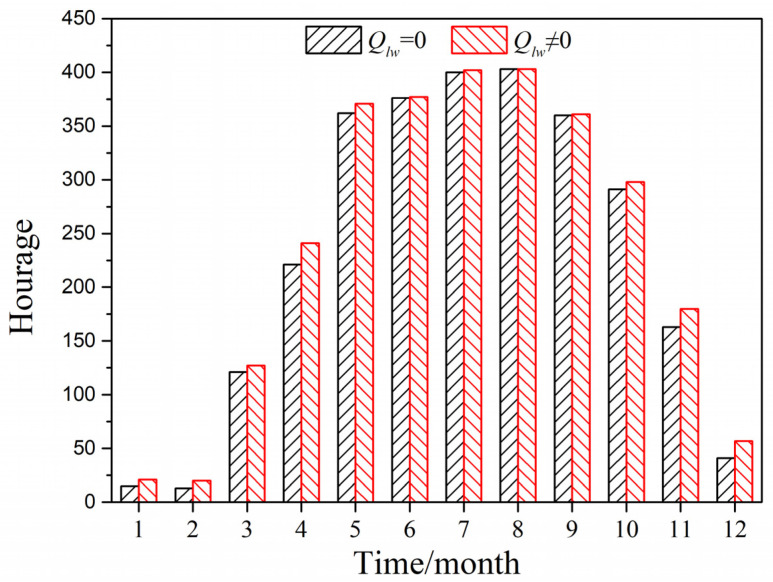
Cumulative hours of greenhouse effect in Guangzhou City.

**Figure 18 entropy-24-01446-f018:**
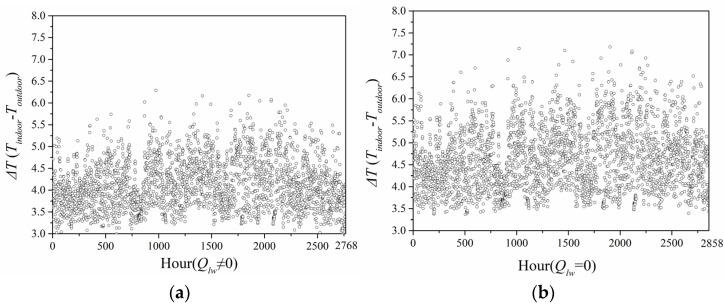
Intensity of greenhouse effect in Guangzhou City: (**a**) considering long-wave radiation; (**b**) neglecting long-wave radiation.

**Table 1 entropy-24-01446-t001:** Components with materials and sizes of the experimental PTH.

Components	Materials and Sizes
Roof	Steel panel (0.5 mm) + polystyrene (75 mm) + steel panel (0.5 mm)
Wall	Steel panel (0.5 mm) + rock wool (50 mm) + steel panel (0.5 mm)
Floor	Plywood (13 mm)
Window	Single-layer glass (3 mm):

**Table 2 entropy-24-01446-t002:** Physical and thermal properties of the materials.

Materials	Density (kg/m^3^)	Specific Heat (J/(kg·K))	Thermal Conductivity (W/(m·K))	Absorptivity	Emissivity
Steel	8000	460	45.28	0.7	0.9
Polystyrene	20	1100	0.035	/	/
Rock wool	71	1100	0.041	/	/
Plywood	521	1630	0.15	/	/
Glass	2500	840	5.9	0.06	0.94

**Table 3 entropy-24-01446-t003:** Details of the sensors utilized in the experiment.

Sensors	Model	Range	Accuracy
Heat flux	JTDL-80	0–2000 W/m^2^	±5%
Sun radiation	9350A	0–2000 W/m^2^	±5%
Wind speed	Testo 480	0.4–50 m/s	±0.2 m/s
Relative humidity	Testo175 H1	0–100% R.H.	±2% RH
Temperature	T-type thermocouple	−200–350 °C	±0.5 °C

**Table 4 entropy-24-01446-t004:** The difference between the two predicted values of the PTH’s exterior-surface temperature in summer.

Temperature Difference (°C)	South	West	East	North	Roof
Average value	1.01	1.87	1.54	0.96	6.76
Max value of daytime	1.91	9.03	12.79	9.36	14.56
Max value of night	0.79	0.79	0.44	0.44	4.25

**Table 5 entropy-24-01446-t005:** The RMSE and CV between the two predicted and the measured exterior-surface temperature in summer.

	South		West		East		North		Roof	
	*Qlw* ≠ 0	*Qlw* = 0	*Qlw* ≠ 0	*Qlw* = 0	*Qlw* ≠ 0	*Qlw* = 0	*Qlw* ≠ 0	*Qlw* = 0	*Qlw* ≠ 0	*Qlw* = 0
RMSE (°C)	1.46	1.58	2.10	4.06	2.53	4.36	2.88	3.45	2.47	6.32
CV (%)	4.73	5.81	6.61	12.78	7.67	13.25	8.79	10.55	6.76	17.32

**Table 6 entropy-24-01446-t006:** The difference between the two predicted values of the PTH’s exterior-surface temperature in winter.

Temperature Difference (°C)	South	West	East	North	Roof
Average value	0.77	0.77	1.56	0.97	5.67
Max value of daytime	1.59	1.66	6.91	2.38	11.88
Max value of night	0.47	0.47	0.54	0.54	4.46

**Table 7 entropy-24-01446-t007:** The RMSE and CV between the two predicted and the measured exterior-surface temperature in winter.

	South		West		East		North		Roof	
	*Qlw* ≠ 0	*Qlw* = 0	*Qlw* ≠ 0	*Qlw* = 0	*Qlw* ≠ 0	*Qlw* = 0	*Qlw* ≠ 0	*Qlw* = 0	*Qlw* ≠ 0	*Qlw* = 0
RMSE (°C)	1.02	1.36	1.02	1.49	2.61	3.55	1.03	1.64	3.88	7.35
CV (%)	7.44	9.92	7.71	11.32	16.29	22.18	7.22	11.53	27.14	51.41

**Table 8 entropy-24-01446-t008:** The difference between the two predicted values of the PTH’s interior-surface temperature in summer.

Temperature Difference (°C)	South	West	East	North	Roof
Average value	0.74	0.81	0.78	0.73	2.50
Max value of daytime	2.67	2.80	3.62	3.32	5.64
Max value of night	0.45	0.46	0.39	0.39	1.62

**Table 9 entropy-24-01446-t009:** The RMSE and CV between the two predicted and the measured interior-surface temperature in summer.

	South		West		East		North		Roof	
	*Qlw* ≠ 0	*Qlw* = 0	*Qlw* ≠ 0	*Qlw* = 0	*Qlw* ≠ 0	*Qlw* = 0	*Qlw* ≠ 0	*Qlw* = 0	*Qlw* ≠ 0	*Qlw* = 0
RMSE (°C)	1.30	1.43	1.40	1.63	1.22	1.57	1.88	1.64	1.93	2.29
CV (%)	4.08	4.52	4.26	4.97	3.84	4.91	5.79	5.04	5.90	6.94

**Table 10 entropy-24-01446-t010:** The difference between the two predicted values of the PTH’s interior-surface temperature in winter.

Temperature Difference (°C)	South	West	East	North	Roof
Average value	0.44	0.45	0.51	0.46	2.38
Max value of daytime	0.95	0.93	1.42	1.00	5.41
Max value of night	0.28	0.28	0.29	0.29	1.47

**Table 11 entropy-24-01446-t011:** The RMSE and CV between the two predicted and the measured interior-surface temperature in winter.

	South		West		East		North		Roof	
	*Qlw* ≠ 0	*Qlw* = 0	*Qlw* ≠ 0	*Qlw* = 0	*Qlw* ≠ 0	*Qlw* = 0	*Qlw* ≠ 0	*Qlw* = 0	*Qlw* ≠ 0	*Qlw* = 0
RMSE (°C)	1.57	1.82	1.40	1.50	1.88	1.83	1.85	1.64	1.92	2.17
CV (%)	10.97	12.73	9.54	10.47	13.09	13.82	13.03	11.53	15.91	20.97

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
