# Peer review of "Theoretical and Experimental Study on the Impact of Long-Wave Radiation on the Greenhouse Effect of a Prefabricated Temporary House"

_entropy, 2022, doi:10.3390/e24101446_

Round 1

Reviewer 1 Report

In this paper, the authors presented an experimental rig of the prefabricated temporary house (PTH), and then they proposed predicted models for thermal environment of the PTH with and without considering long-wave radiation. Next, calculated results were then compared with the experimental results to study the influence of long-wave radiation on the predicted characteristic temperature of the PTH. Furthermore, the authors calculated the cumulative annual hours and the intensity of the green-house effect of four different climate cities (Harbin, Beijing, Chengdu, Guangzhou) with predicted models. The work in this study is meaningful and full of scientific value. It can be considered to be published after some revisions with comments below:

1.     Please add some latest literature reviews on the effect of long-wave radiation on the thermal environment of the PTH.

2.     Please give the more specific descriptions on distribution of the temperature sensors.

3.     Please indicate why the temperatures are recorded at an interval of 5 min.

4.     Please give out the detailed meanings of the key symbols for the equations in the Section 3.

5.     If possible, please give more detailed analysis for the results obtained in the Section 4.

Reviewer 2 Report

The review of a manuscript titled " Theoretical and Experimental Study for the Impact of Long-wave Radiation on the Green-house Effect of a Prefabricated Temporary House" by Wen and Long

The following aspects have to be clarified before the manuscript can be accepted.

1.       All the constants used in equation (2) have to be clearly stated. Only properties of the materials are given in table 2.

2.       How are thermocouples fixed temperature measurement of exterior surfaces? How many thermocouples are fixed on each side?

3.       Since this a transient modelling how did the authors assume initial conditions? What are these conditions?

4.       In table 4 and 6, are temperature differences with or without long-wave radiation modeling?

5.       What is the reason for CV to be so large in winter compared to summer for exterior temperature measurement? Is this because you did not consider any specific component of radiation or problem of temperature measurement during winter?

6.       What is the reason for consistent phase shift in measured and calculated temperatures for indoor temperatures? 

Round 2

Reviewer 2 Report

The authors have not addressed comment number 2. I have intended that all the constants in the equation 2 should be specified like overall heat transfer coefficient etc. Table 4 and 6, error are with long thermal radiation? Or without thermal tradition? Please be specific.
